# The impact of IOM recommendations on gestational weight gain among US women: An analysis of birth records during 2011–2019

**Vidhura S. Tennekoon** [ORCID] *

Department of Economics, School of Liberal Arts, Indiana University Purdue University Indianapolis (IUPUI), Indianapolis, Indiana, United States of America

* vtenneko@iupui.edu

## Abstract

The prevailing guidelines of the Institute of Medicine (IOM) of United States on gestational weight gain (GWG) are based on women's prepregnancy body mass index (BMI) categories. Previous research has shown that the guidelines issued in 1990 and revised in 2009 had no effect. We investigate the effectiveness of new guidelines issued in 2009 analyzing the records of all singleton births in the U.S. during 2011–2019 (34.0 million observations). We use the discontinuity in recommended guidelines at the threshold values of BMI categories in a regression discontinuity (RD) research design to investigate the effect of IOM guidelines on GWG. We also use an RD analysis in a difference in difference (DID) framework where we compare the effect on women who had any prenatal care to others who did not receive prenatal care. The naïve RD estimator predicts an effect in the expected direction at the threshold BMI values of 18.5 and 25.0 but not at 30.0. After the DID based correction, the RD analyses show that the GWG, measured in kg, drop at the BMI values of 18.5, 25.0 and 30.0 by 0.189 [CI: 0.341, 0.037], 0.085 [CI: 0.179, -0.009] and 0.200 [CI: 0.328, 0.072] respectively when the midpoint of the recommended range in kg drops by 1.5, 4.5 and 2.25. This implies a responsiveness of 12.6%, 1.9% and 8.9% respectively to changes in guidelines at these BMI values. The findings show that the national guidelines have induced some behavioral changes among US women during their pregnancy resulting in a change in GWG in the expected direction. However, the magnitude of the change has not been large compared to the expectations, implying that the existing mechanisms to implement these guidelines have not been sufficiently strong.

## Introduction

Improving the well-being of mothers, infants, and children is an important public health goal. Over the past half century, the guidance provided to women before, during, and after pregnancy to achieve this goal has changed dynamically when public health trends change. A positive relationship between gestational weight gain (GWG) and birth weight was identified in a report *Maternal Nutrition and the Course of Pregnancy* published by the Committee on

**Data Availability Statement:** All analyses presented here are based on 2011-2019 public use US birth data files available at the Vital Statistics Online Data Portal of the National Center for Health

Statistics. Interested readers may use following link to download data: https://www.cdc.gov/nchs/data_access/vitalstatsonline.htm.

**Funding:** This work was supported by two grants to VST; the Enhanced Mentoring Program with Opportunities for Ways to Excel in Research (EMPOWER) and Emergency Equity Fund for Research, both from Indiana University. The funders had no role in study design, data collection and analysis, decision to publish, or preparation of the manuscript.

**Competing interests:** The authors have declared that no competing interests exist.

Maternal Nutrition of National Research Council (NRC) in 1970 which recommended an average GWG of 20–25 pounds (9.1–11.3 kg) against the previous recommendation of 10–14 pounds (4.5–6.4 kg) [1]. During the 20 years since publishing this report, several additional studies enhanced the existing knowledge on the effect of nutrition during pregnancy and the perinatal period [2, 3]. During the same period, the Supplemental Food Program for Women, Infants, and Children (WIC) was established in the U.S. Department of Agriculture as a permanent program with the aim of improving the health of pregnant mothers, infants, and children. During the period from 1971 to 1980, mean birth weight increased by approximately 60g for whites and 30g for blacks, prevalence of low birth weight was reduced by about 20% for whites and 7% for blacks, and the prevalence of high birth weight increased by 30% for whites and by 15% for blacks, probably due to increased prepregnancy weight, increased height, decreased smoking during pregnancy, increased participation in the WIC program, and earlier prenatal care [4]. These changes in birth outcomes motivated the revised IOM guidelines introduced in 1990, which were based on BMI categories (underweight, normal weight, overweight, and obese) unlike the then existed uniform guidelines (see Table 1).

Trends in pregnancy and birth related outcomes continued to change during the 19 years after introducing the 1990 guidelines. The population of U.S. women of childbearing age became more diverse while the average age of a pregnant woman increased. The percentage of women who were overweight or obese when entering pregnancy increased gradually. These trends and newer research prompted to revise the 1990 guidelines and issue new guidelines in 2009. The 2009 IOM guidelines for singleton births which are still in use are presented in Table 1, together with 1970 and 1990 guidelines [5].

The new guidelines also were based on BMI categories, like the previous guidelines. The main difference between the guidelines issued in 1990 and 2009 was on how BMI categories were defined. The cutoff points used to separate the four BMI categories in 1990 corresponded to 90%, 120%, and 135% of the weight-for-height standards prepared by the Metropolitan Life Insurance Company and most widely used in the US at that time [6]. A few years after issuing the 1990 IOM recommendations, the World Health Organization (WHO) introduced a different set of cutoff points to define BMI categories which were subsequently endorsed by the National Institutes of Health [7, 8]. The BMI categories of WHO were widely used in the US and internationally by the time when 2009 IOM recommendations were prepared, and the new guidelines were based on those categories. The recommendation for underweight, normal weight and overweight categories in 2009 were the same as those prevailed since 1990 but the

**Table 1. IOM guidelines for women with singleton fetuses since 1970.**

| Prepregnancy BMI (kg/m$^2$) | Recommended GWG in lbs. | Recommended GWG in kg |
|---|---|---|
| **During 1970–2009** | | |
| All BMI categories | 20–25 | 9.1–11.3 |
| **During 1990–2009** | | |
| $\leq$ 19.7 (underweight) | 28–40 | 12.5–18.0 |
| 19.8–26.0 (normal weight) | 25–35 | 11.5–16.0 |
| 26.1–30.0 (overweight) | 15–25 | 7.0–11.5 |
| $\geq$ 29.1 (obese) | $\geq$ 15.0 | $\geq$ 6.8 |
| **From 2009** | | |
| $\leq$ 18.4 (underweight) | 28–40 | 12.5–18.0 |
| 18.5–24.9 (normal weight) | 25–35 | 11.5–16.0 |
| 25.0–29.9 (overweight) | 15–25 | 7.0–11.5 |
| $\geq$ 30.0 (obese) | 11–20 | 5.0–9.0 |

cutoff point between the first two categories was shifted from 19.8 to 18.5 and the cutoff point separating the next two categories was shifted from 26.0 to 25.0. The lower limit of the obese category was moved from 29.0 to 30.0. In addition, the recommended range of GWG for that category was changed to 11–20 pounds from the previous recommendation of over 15 pounds with no upper limit.

When the 2009 IOM guidelines were prepared, the effect of GWG on a broad spectrum of outcomes including pregnancy and birth outcomes, neonatal outcomes, postpartum outcomes as well as the long-term health consequences affecting both mother and the child were considered. Increases in body weight during pregnancy involve both maternal components including RBC mass, body water, fat, and uterine and breast tissue; and the products of conception, the fetus, placenta, and amniotic fluid [9]. Maternal components contribute to approximately 65% of GWG; the rest is products of conception [10]. Changes in body weight, fat mass, and fat-free mass during pregnancy reflect changes in maternal nutritional status throughout this nutritionally challenging period [10, 11]. Maternal prepregnancy weight and GWG are strong predictors of intrauterine growth and birth weight which influences the growth and survival of an infant [12]. The multifaceted process of transmitting a woman's dietary intake during pregnancy towards the birth weight of an infant is moderated through GWG patterns [11].

The effectiveness of IOM guidelines depends crucially on the adherence to these guidelines by pregnant women. Recognizing this, the IOM/NRC committee recommended ways to promote adoption of these GWG guidelines through consumer education, strategies to assist practitioners, and public health strategies [5]. In 2013, IOM and NRC published implementation guidelines for their 2009 GWG recommendations which prescribed practitioners to record prepregnancy BMI, chart weight gain throughout pregnancy, share the results with the patient and provide counselling, among other things [13].

The IOM guidelines introduced in 2009 have been adopted by many other countries even though the report explicitly mentioned that they are intended for use among women in the United States. A comparative study of national GWG guidelines published in 2012 found that the guidelines of about half of the countries they studied were similar to 2009 IOM guidelines [14]. Benefits of following these guidelines are documented for US women and infants as well as for populations outside the US [15–19]. Despite the wider acceptance of these guidelines among researchers and medical practitioners, its acceptance within the targeted audience, the pregnant women, has not been satisfactory as indicated by the low rate of compliance with these guidelines. A meta-analysis based on more than 1.3 million women in 23 studies found that only 30% of women in the respective samples had a GWG within the ranges of IOM recommendations [16]. Data from US birth records confirm this number. A part of the problem could be due to weaknesses in implementing these guidelines. Some researchers have pointed out that the existing system of healthcare delivery does not always adhere to evidence based best practice guidelines [20–22]. According to one study, healthcare providers typically provide GWG information early in pregnancy, but not again unless there is a concern [23]. In addition, a review of recent literature on GWG highlights many inconsistencies in the interpretation and application of 2009 IOM guidelines [24]. Even if the existing healthcare delivery system is efficient, inducing behavioral changes that would impact GWG through guidelines alone is not an easy task. At the first place, behaviors of a women influencing her GWG such as calory intake and physical exercise is a utility maximizing combination of choices from a behavioral economic perspective and may not necessarily reflect a knowledge gap. A recommendation, unlike a financial incentive or a penalty, can alter behaviors only when there is a knowledge gap due to an information asymmetry. Therefore, the adherence to a guideline is not guaranteed even when a woman is fully aware of the evidence-based benefits.

In this study, we use US birth records during 2001–2019 to investigate the effectiveness of 2009 IOM guidelines. The only existing work with a comprehensive evaluation of IOM guidelines on GWG is Hamad, Cohen and Rehkopf (2016), HCR hereafter [25]. Using a quasi-experimental study HCR show that 1990 IOM guidelines on GWG had no effect on pregnant women. No study has evaluated the effectiveness of IOM guidelines after the 2009 revision in a broader way as HCR analyzed the 1990 guidelines. That study, however, is based on a sample of 4,173 women (7,442 pregnancies) from the National Longitudinal Survey of Youth who self-reported their GWG for pregnancies during 1979–2000. Their main identification strategy is based on the discontinuity in recommended GWG resulted from the introduction of new guidelines in July 1991. In other words, they investigate whether there is any discrete change in average GWG before and after implementing 1990 guidelines in a regression discontinuity (RD) framework. The authors also use a difference-in-difference (DID) strategy which also relies on the difference before and after implementing the guidelines. A limitation of this approach is that the identification strategy fails if the policy change diffused to practitioners and patients slowly and the new recommendation was not implemented across US at the same time in July 1991.

One of the key observations when the IOM guidelines were introduced in 2009 was that the data sources then available were inadequate for studying national trends in GWG. The IOM/NRC committee recommended all states to adopt the revised version of the birth certificate, which includes fields for maternal prepregnancy weight, height, delivery weight and weight at delivery, and gestational age. This recommendation was followed subsequently, and the details are available for all births registered in the US since 2011. We use records of all singleton births in the US during 2011–2019 for our analysis. This amounts to 34.0 million observations representing 10.3% of the current US population. We utilize this data to see a more accurate and detailed picture of the effect of 2009 IOM guidelines on GWG. This is the first comprehensive study to investigate the effectiveness of prevailing IOM guidelines on GWG of US women. Unlike an analysis based on a representative sample survey, our analysis is free from sampling bias for the population it represents, and the large sample size allows us to accurately identify even a smaller effect. Ours also is a RD based identification strategy as HCR. However, we use discontinuity in recommended GWG at the cutoffs separating the BMI categories, rather than the discontinuity at the date of implementing these guidelines.

## Materials and methods

### Dataset

All analyses presented here are based on 2011–2019 public use US birth data files available at the Vital Statistics Online Data Portal of the National Center for Health Statistics. The combined full dataset includes the details contained in 35,229,670 birth records representing 10.7% of 2019 US population. Our analyses are based on 34,021,301 singleton births which represent 96.6% of these records.

### Measures

The two measures of interest in our study are prepregnancy BMI and GWG. Both these measures have been derived using the information about maternal prepregnancy weight, delivery weight, and height. US national standard birth certificate did not include information on maternal prepregnancy weight and height prior to the 2003 revision [26]. It was recommended during the 2003 revision of US national standard birth certificate that prepregnancy weight (in pounds) and height (in feet and inches) be collected at the time of delivery. The exact question asked about prepregnancy weight is "*What was your prepregnancy weight, that is, your weight*

*immediately before you became pregnant with this child*?". Information on prepregnancy weight and height are collected from the self-reported Mother's Worksheet [27]. Weight at delivery is collected directly from the medical record using the Facility Worksheet [28]. GWG was derived from mother's prepregnancy weight and mother's weight at delivery and converted to grams. Prepregnancy BMI was derived from mother's prepregnancy weight and height as, mother's pre-pregnancy weight in pounds x 703/ (mother's height in inches)² rounded to one decimal place [29]. Other confounding variables we use in our study also were based on self-reported information in Mother's Worksheet.

## The RD approach

The 2009 IOM guidelines have three thresholds at the BMI values of 18.5, 25.0 and 30.0 when BMI categories change (Table 1). At each of these threshold values the upper bound of the recommended range in kg drops by 2.0, 4.5 and 2.5; lower bound drops by 1.0, 4.5 and 2.0; and the midpoint drops by 1.5, 4.5 and 2.25 as shown in Fig 1. The idea behind the RD approach is that the relationship between GWG (outcome variable) and prepregnancy BMI value (rating variable) could potentially change at the BMI values equal to 18.5, 25.0 and 30.0 (the threshold values) because of IOM guidelines but for no other reason. Therefore, any changes in GWG at one of these threshold values can be interpreted as the causal effect of IOM guidelines in GWG at that value.

The first step of any RD analysis is the visual inspection of data to identify the nature of the relationship between the outcome variable and the rating variable including any clearly visible shift in the value of the outcome at the threshold values. We noticed a clear and consistent negative relationship between the two variables; average GWG drops with prepregnancy BMI in its entire range. Moreover, this relationship was non-linear. However, we did not observe any

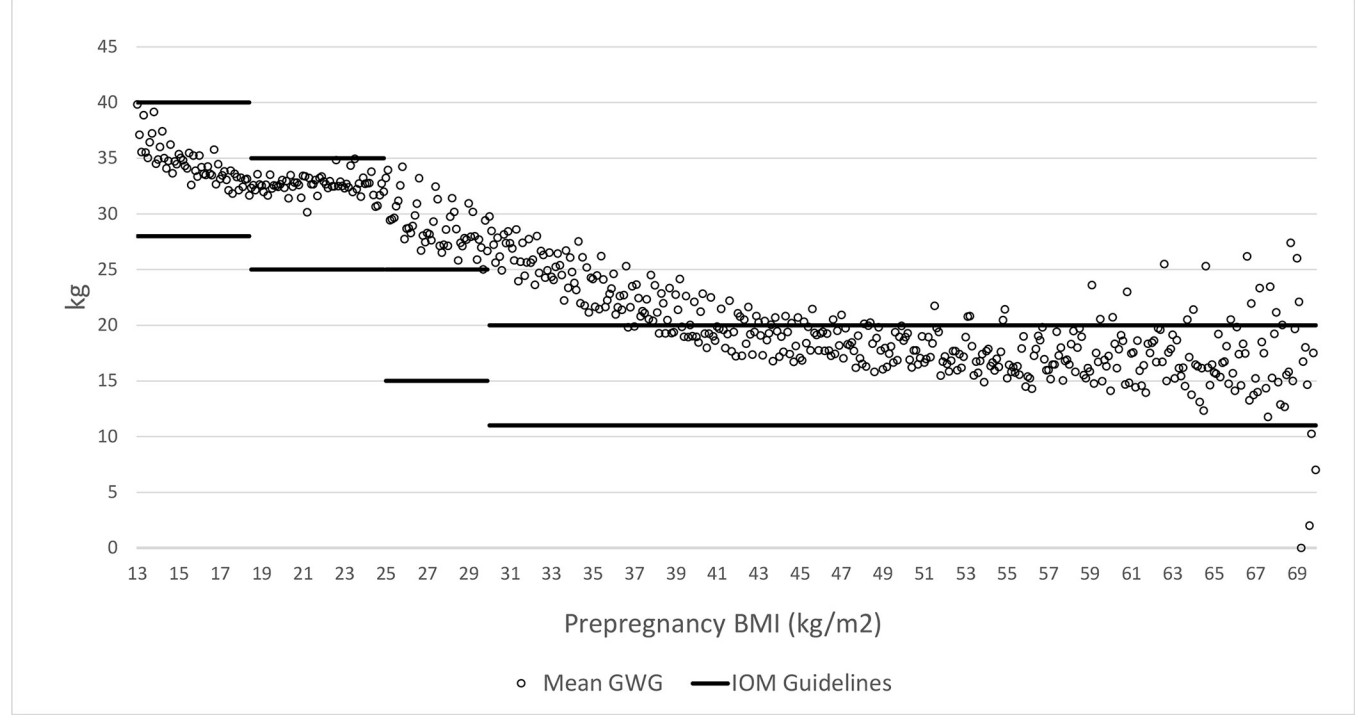

**Fig 1. Average GWG at each value of prepregnancy BMI and IOM guidelines.**

**Table 2. Change in GWG at the cutoffs between BMI categories.**

| Cutoff value (kg/m$^2$) | Mean GWG below cutoff (kg)[a] | Mean GWG above cutoff (kg)[b] | Mean difference (kg)[c] |
|---|---|---|---|
| 18.5 | 14.972 | 14.826 | -0.146*** (16.650) |
| 25.0 | 14.610 | 14.515 | -0.094*** (14.831) |
| 30.0 | 12.817 | 12.497 | -0.320*** (33.643) |

Notes

[a] Based on a range within 1 unit below each cutoff.

[b] Based on a range within 1 unit above each cutoff including the cutoff value.

[c] t-statistics are within parenthesis

* p<0.1

**p<0.05

***p<0.01.

clearly visible shift in outcome at any of the three threshold values. Therefore, we moved to test whether the mean of the outcome is statistically different before and after the cutoff value in its local neighborhood. For that purpose, the mean GWG within a unit (in kg/m$^2$) below each BMI cutoff was t-tested against the mean GWG within a unit above each BMI cutoff. The results presented in Table 2 indicate that the means are statistically different, implying a possible change at the cutoffs in response to the guidelines. The mean differences were -0.146, -0.094 and -0.350kg, respectively, around the cutoff values of 18.5, 25.0 and 30.0. However, these differences may represent the general trend in the relationship between the rating variable and the outcome variable, not necessarily a response to IOM guidelines. It also may represent differences between the two groups due to confounders.

The effect of IOM recommendations at the BMI cutoff values can be identified precisely using a regression model which identifies the nature of the relationship between the two variables as well as any change in that relationship at the threshold values. This approach also allows to measure the statistical accuracy of the estimated effect. In order to identify the effect of IOM recommendations on GWG at a given threshold value, we used data belonged to each couple of two adjacent categories. For example, the data belonged to underweight and normal weight categories were used to measure the effect at the BMI value of 18.5. This allows to model the relationship between GWG and BMI more precisely which we assumed to be quadratic in each of the three local regions based on the non-linear relationship between the rating variable (BMI value) and the outcome variable (GWG) we noticed in Fig 1. In the regression estimates restricted to a specific range of BMI values, the assumed quadratic relationship potentially breaks only at one value. Each regression model we estimated using OLS takes the following form where $D_i$ is a dummy variable which takes the value 0 at the points below the threshold and 1 at the values above the threshold.

$$GWG_i = \alpha + \delta D_i + \beta_1 BMI_i + \beta_2 BMI_i^2 + \gamma X_i + \varepsilon_i \qquad (1)$$

Our parameter of interest is $\delta$, the discontinuity at the cutoff point in response to each discrete jump in IOM guidelines. $X_i$ is a vector of additional control variables.

## The RD approach in a DID setting

In above quasi-experimental research design, we assumed there is no reason for the relationship between GWG and prepregnancy BMI to break at the threshold values except for the change in IOM guidelines. These exact thresholds determine the BMI categories of a woman. If the identity with a specific BMI category of a woman induces any psychological effect and

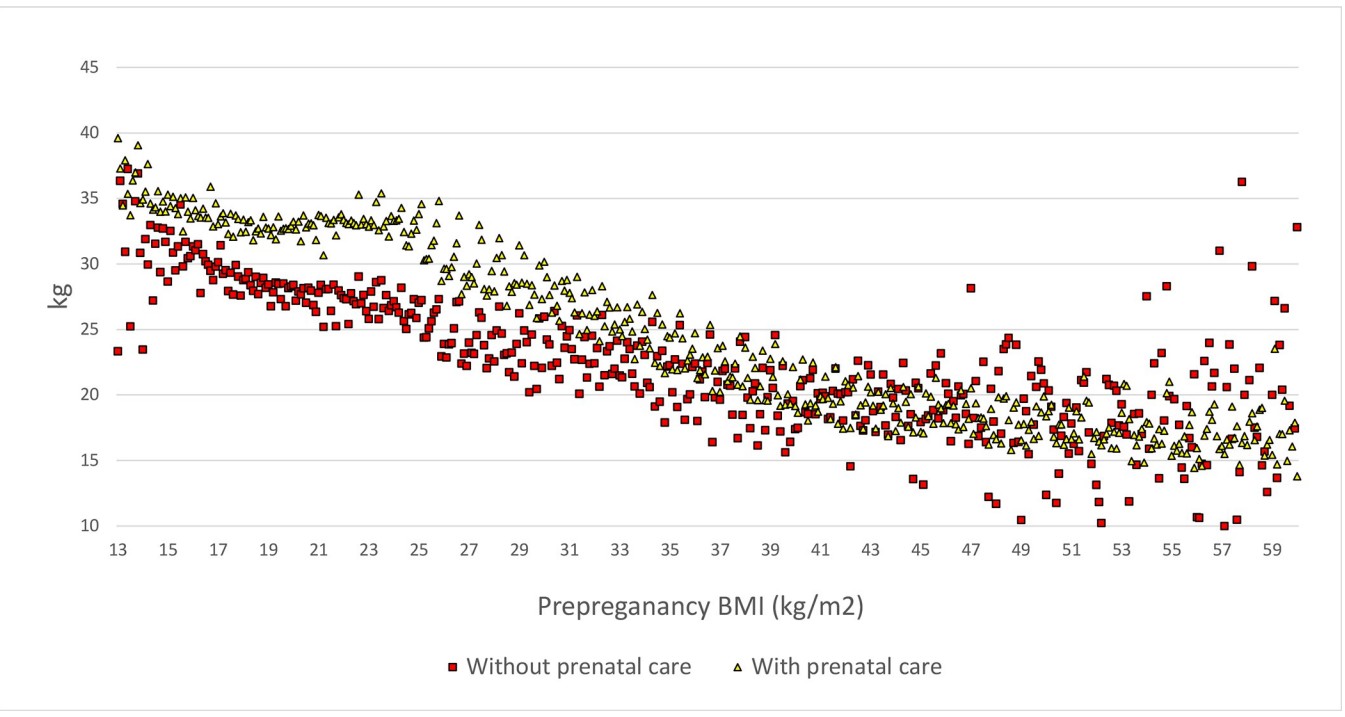

**Fig 2. GWG and prepregnancy BMI with and without prenatal care.**

thereby a behavioral change which affects GWG, our identifying assumption fails. A modified RD research design can circumvent this problem. In this research design, we make the additional assumption that a woman is unaware of IOM guidelines if she never receives prenatal care during her pregnancy and therefore any estimated effect using our naïve RD design on this group represents a placebo effect. The true effect of IOM guidelines on GWG net of this placebo effect can be found by the difference in RD estimates on the group of women who received any prenatal care and others who did not. An identified weakness in HCR, which we also need to address, is the measurement error in self-reported anthropometric measurements [30, 31]. If any measurement error in self-reported anthropometric measurements were due to social desirability bias or recall bias and that bias is uncorrelated with prenatal care this DID-based RD approach also eliminates that bias by differencing [32]. The regression equation we estimated was the following and the parameter of interest was $\delta$.

$$GWG_i = \alpha + \beta_1 D_i + \beta_2 PC_i + \delta D_i * PC_i + \beta_3 BMI_i + \beta_4 BMI_i^2 + \gamma_1 \boldsymbol{X}_i + \beta_5 BMI_i * PC_i \\ + \beta_6 BMI_i^2 * PC_i + \gamma_2 \boldsymbol{X}_i * PC_i + \boldsymbol{\varepsilon}_i \qquad (2)$$

For identification, we assumed that each of the two groups have quadratic relationship between GWG and prepregnancy BMI, an assumption justified by Fig 2. We did not assume similarity of the two groups or a parallel trend.

## Control variables

We have included age, age squared, education, marital status, race categories, smoking during pregnancy (as a proxy for risky health behaviors) and WIC participation (as a proxy for low household income) as controls. Sensitivity of results to each of these covariates was tested.

## Results

### Sample characteristics

The mean prepregnancy BMI of women was 26.7 in the full sample, a value falls within the overweight category. Only 3.3% of women were in underweight category while 41.3% were in normal weight category, 24.0% were overweight, and 31.4% were obese. Overall, 29.8% of women had a GWG within the recommended range, 51.0% were above and 20.1% were below the recommended range. Rate of compliance with IOM guidelines decreased when prepregnancy BMI increased. Other differences and similarities between BMI categories are summarized in Table 3.

### Regression discontinuity analyses

The naïve RD estimator predicts an effect in the expected direction at the BMI values of 18.5 and 25.0 but not at 30.0. After the DID based correction, a drop in GWG was noticed at each of the three threshold values (Table 4). The midpoint of the recommended range in kg drops by 1.5, 4.5 and 2.25 respectively at the threshold values of 18.5, 25.0 and 30. Therefore, our estimates imply a responsiveness of 12.6%, 1.9% and 8.9%, respectively, at these BMI values in response to changes in guidelines. The results were robust (qualitatively similar) to the exclusion of any or all the covariates.

## Discussion

Maternal prepregnancy BMI and GWG independently increase various health risks [33]. During the prenatal period, pregnant women are willing to make lifestyle changes for the benefit of their offspring and they are in close contact with their healthcare providers through routine

**Table 3. Sample characteristics.**

| Variable (mean) | Under weight | Normal weight | Overweight | Obese |
|---|---|---|---|---|
| GWG (kg) | 15.13 | 14.83 | 13.64 | 10.55 |
| Age (years) | 26.28 | 28.33 | 28.69 | 28.60 |
| *Education* | | | | |
| 8th grade or less | 0.029 | 0.030 | 0.045 | 0.039 |
| 9th through 12th grade | 0.158 | 0.103 | 0.110 | 0.116 |
| High school graduate or GED | 0.290 | 0.224 | 0.251 | 0.289 |
| Some college credit | 0.191 | 0.182 | 0.210 | 0.240 |
| Associate degree | 0.059 | 0.074 | 0.084 | 0.088 |
| Bachelor's degree | 0.164 | 0.231 | 0.188 | 0.143 |
| Master's degree | 0.072 | 0.109 | 0.082 | 0.057 |
| Doctorate or professional degree | 0.025 | 0.036 | 0.020 | 0.012 |
| Married | 0.495 | 0.606 | 0.570 | 0.534 |
| *Race* | | | | |
| White | 0.687 | 0.761 | 0.751 | 0.716 |
| Black | 0.140 | 0.118 | 0.159 | 0.207 |
| American Indian and Alaska Native | 0.007 | 0.007 | 0.010 | 0.013 |
| Asian | 0.140 | 0.089 | 0.053 | 0.031 |
| Native Hawaiian and other Pacific Islanders | 0.002 | 0.002 | 0.003 | 0.005 |
| Smoked during pregnancy | 0.129 | 0.071 | 0.071 | 0.222 |
| WIC participation | 0.451 | 0.350 | 0.416 | 0.407 |
| Number of observations | 1,126,929 | 14,059,319 | 8,156,377 | 10,678,676 |

 

**Table 4. Discontinuity at threshold (kg).**

| Threshold | RD Estimates [CI] | RD/DID Estimates [CI] |
|---|---|---|
| 18.5 | -0.244*** [-0.265, -0.224] | -0.189** [-0.341, -0.037] |
| 25.0 | -0.020*** [-0.031, -0.009] | -0.085* [-0.179, 0.009] |
| 30.0 | 0.218*** [0.203, 0.233] | -0.200***[-0.328, -0.072] |

Note

* p<0.1

**p<0.05

***p<0.01

prenatal care visits [34]. Lifestyle interventions initiated during this period may yield lasting positive benefits for both mother and the child [35, 36]. The IOM guidelines are useful for monitoring GWG and intervene so that the risks of several adverse outcomes such as cesarean delivery, macrosomia, preterm birth and LBW could be minimized [37]. In this study, we examined the effect of 2009 IOM recommendations on GWG using 2011–2019 US birth records. The dataset used has over 34 million observations and includes 10.3% of US population in 2019. Our estimates show that the overall responsiveness to the guidelines is not very high. At the prepregnancy BMI value of 25.0, the threshold value which separates the overweight and normal weight categories, the responsiveness is only 1.9%.

The only previous study (HCR) which exclusively investigates the effect of IOM recommendations on GWG is based on a survey data sample and covers a period before the 2009 revision of guidelines but uses quasi experimental research designs closely related to ours [25]. They found no statistically significant change in GWG both when the guidelines were revised in 1990 and at the BMI value which separates overweight and normal weight categories. We found statistically significant effects at each of the threshold values separating the BMI categories. While the results can be different due to the time gap between the datasets used in two studies, our results can easily be reconciled with HCR using the differences in research designs and the number of observations in each dataset.

Authors of HCR explain their null findings arguing that the recommendations, probably, were not adequately disseminated to patients and providers, citing prior research in support of their argument [38]. This argument is also supported by the facts that implementation guidelines for 2009 recommendations were not published until 2013 and the American College of Obstetricians and Gynecologists delayed endorsing these guidelines until the same year [13, 39]. Therefore, the observation that the GWG did not change immediately following the 1990 revision is no surprise. The second null finding in HCR is based on the discontinuity at the threshold value between normal and overweight categories. While we found a significant effect at that value, the magnitude of the effect was only 1.9% of the change in recommended range of guidelines (0.085kg). This effect was too small to be detected using a smaller sample size. While responsiveness to the guidelines based on our RD estimates was only 1.9% at the BMI value of 25.0, it was 12.6% at 30.0 and 8.9% at 18.5. This implies that pregnant women have been less likely to respond to sharp changes in recommended guidelines as at the BMI value of 25.0. Therefore, a gradual decline in recommended range of GWG with prepregnancy BMI could be more effective in improving compliance.

While our results show a statistically significant effect of guidelines, that effect is far less than the intended effect. Addressing this gap requires clearly identifying why the gap exists. It has been suggested that this gap could be a result of limitations in the implementation procedure and if that is the case the appropriate solution is motivating healthcare providers through

 

some action. However, knowledge about a guideline alone does not guarantee compliance, a best example is vaccine hesitancy. A woman's prepregnancy BMI as well as the GWG is a result of her lifestyle choices though other factors such as genetics also play a role. The only tool available for physicians and other healthcare provides to implement IOM guidelines is patient education but lifestyle choices are hard to change through education alone and may require other types of interventions. A survey of pregnant women may help to understand the role of various factors causing the gap we identify here and respond as appropriate. Only through such a well-designed survey we can identify whether most women are unaware of the guidelines, they are aware but rely more on misinformation, or they simply ignore the guidelines. If most women are aware of the guidelines and they ignore those, a survey will also help to identify the reasons underlying such behavior.

Accuracy of our estimates are limited by measurement error in data. While the delivery weight was based on clinical records, prepregnancy weight was based on self-reports which were subject to various sources of bias. Measuring the BMI of a women accurately at the time of conception is challenged by the fact that over a half of pregnancies in the US are unplanned [40]. Self-reported weight is significantly underreported and the extent of underreporting increases with BMI, even though some research finds a higher level of agreement between self-reported and clinical measurements [41–43]. Another clearly observable tendency was reporting 'round' numbers, multiples of five, for example [44]. Since GWG was derived using the delivery weight and prepregnancy weight, this 'heaping issue' may have added systemic measurement error on GWG. The measurement error in prepregnancy weight could also propagate to prepregnancy BMI and thus distort the relationship between observed prepregnancy BMI and GWG. If any bias due to this distortion has been similar among the two groups who received prenatal care and who did not, our DID strategy may have eliminated this bias but not if this assumption was violated.

While we found an effect of IOM guidelines on GWG, the effect was not very large compared to the expectations. One of the potential reasons behind this observation is limited compliance; less than 30% of women in our dataset has followed the recommendations. Future research should focus on investigating the cost of noncompliance so that the cost of any intervention to improve compliance can be justified.

## Acknowledgments

We thank Anne Beeson Royalty, Mark Ottoni-Wilhelm, the academic editor and two anonymous referees for many helpful comments.

## Author Contributions

**Conceptualization:** Vidhura S. Tennekoon.

**Data curation:** Vidhura S. Tennekoon.

**Formal analysis:** Vidhura S. Tennekoon.

**Funding acquisition:** Vidhura S. Tennekoon.

**Investigation:** Vidhura S. Tennekoon.

**Methodology:** Vidhura S. Tennekoon.

**Project administration:** Vidhura S. Tennekoon.

**Resources:** Vidhura S. Tennekoon.

**Software:** Vidhura S. Tennekoon.

**Validation:** Vidhura S. Tennekoon.

**Visualization:** Vidhura S. Tennekoon.

**Writing – original draft:** Vidhura S. Tennekoon.

**Writing – review & editing:** Vidhura S. Tennekoon.

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
