## [Decision Letter · Decision Letter 0]

12 May 2022

PGPH-D-22-00245

The impact of IOM recommendations on gestational weight gain among US women: An analysis of birth records during 2011-2019.

Dear Dr. Tennekoon,

Thank you for submitting your manuscript to PLOS Global Public Health. After careful consideration, we feel that it has merit but does not fully meet PLOS Global Public Health’s publication criteria as it currently stands. Therefore, we invite you to submit a revised version of the manuscript that addresses the points raised during the review process.

We kindly recommend:

consider expanding the discussion on the implications of the results;suggesting some recommendations that the authors think would improve the situation, as in line 333;placing the limitations (beginning in line 335) in a separate paragraph.

Please submit your revised manuscript by . If you will need more time than this to complete your revisions, please reply to this message or contact the journal office at globalpubhealth@plos.org. Please include the following items when submitting your revised manuscript:

We look forward to receiving your revised manuscript.

Kind regards,

Hanna Nalecz, Ph.D.

Academic Editor

Journal Requirements:

2. Please update your Competing Interests statement. If you have no competing interests to declare, please state: “The authors have declared that no competing interests exist.”

Additional Editor Comments (if provided):

Reviewers' comments:

Reviewer's Responses to Questions

**Comments to the Author**

1. Does this manuscript meet PLOS Global Public Health’s publication criteria? Is the manuscript technically sound, and do the data support the conclusions? The manuscript must describe methodologically and ethically rigorous research with conclusions that are appropriately drawn based on the data presented.

Reviewer #1: Yes

Reviewer #2: Yes

2. Has the statistical analysis been performed appropriately and rigorously?

Reviewer #1: Yes

Reviewer #2: Yes

3. Have the authors made all data underlying the findings in their manuscript fully available (please refer to the Data Availability Statement at the start of the manuscript PDF file)?

Reviewer #1: Yes

Reviewer #2: Yes

4. Is the manuscript presented in an intelligible fashion and written in standard English?

Reviewer #1: Yes

Reviewer #2: Yes

5. Review Comments to the Author

Reviewer #1: Obesity is a form of world pandemic. Having that in mind, I strongly agree with subject that author has elaborated in this article. Especially if you combine obesity with pregnancy and the unwanted outcomes as a result. The author very meticulously describes and elaborates the changes and effects of such guidelines in everyday practice, coming to interesting conclusions through detailed statistical analyses. As such I think that author has meet PLOS Global Public Health’s publication criteria.

Reviewer #2: Authors

The study is well written and addresses an important issue, investigating the effectiveness of new IOM guidelines and analysing the records of all singleton births in the U.S. from 2011 to 2019 (34.0 million observations).

Title & Abstract

You provided a balanced and informative summary of what the study entails and what you did and found in the abstract.

Introduction

The introduction section provided the scientific background and rationale for the study, stating study objectives.

Methods

The methodology was concise and appropriate.

Discussion and conclusions:

My main comment in the method and conclusion sections is that I would have like to see more discussion on the implication of the findings and some recommendations, which I believe would add to the manuscript.

6. PLOS authors have the option to publish the peer review history of their article (what does this mean?). If published, this will include your full peer review and any attached files.

**Do you want your identity to be public for this peer review?** For information about this choice, including consent withdrawal, please see our Privacy Policy.

Reviewer #1: No

Reviewer #2: No

---

## [Editor Report · Decision Letter 1]

27 Jun 2022

The impact of IOM recommendations on gestational weight gain among US women: An analysis of birth records during 2011-2019.

PGPH-D-22-00245R1

Dear Dr. Tennekoon,

We are pleased to inform you that your manuscript 'The impact of IOM recommendations on gestational weight gain among US women: An analysis of birth records during 2011-2019.' has been provisionally accepted for publication in PLOS Global Public Health.

Best regards,

Hanna Nalecz, Ph.D.

Academic Editor